# Assessing soil-transmitted helminths and *Schistosoma mansoni* infections using parasitological indicators after seven years of preventive chemotherapy among school-age children in Mizan-Aman town

Mitiku Abera[1]*, Tariku Belay[2], Daniel Emana[2], Zeleke Mekonnen[2]

**1** Department of Medical Laboratory Sciences, College of Medicine and Health Sciences, Mizan-Tepi University, Mizan-Aman, Ethiopia, **2** School of Medical Laboratory Sciences, Institute of Health, Jimma University, Jimma, Ethiopia

* mitikuaberad@gmail.com

## Abstract

### Background

Soil-transmitted helminthiasis and Schistosomiasis are major public health problems mainly among school-age children. Despite the seven years of implementing elimination program, the ongoing prevalence and intensity of both diseases have not been assessed in the study area. Hence, this study aimed to determine the parasitological indicator (prevalence and intensity) of soil-transmitted helminths and *Schistosoma mansoni* infections after seven years of preventive chemotherapy among school-age children in Mizan-Aman town.

### Methods

A cross-sectional study was conducted among 615 school-age children from January to February 2022 in Mizan-Aman Town. Study participants were selected using a systematic random sampling technique. Sociodemographic and associated factors were collected using a structured questionnaire. The stool samples were collected and processed using the double-slide Kato-Katz technique. The parasitological indicator was evaluated based on the thresholds set by the elimination program.

### Results

The overall prevalence of soil-transmitted helminths infection was 50.7% (95% CI: 46.8-54.8); with 2.4% moderate and heavy intensity, and *Schistosoma mansoni* infection prevalence was 25.4% (95% CI: 22.1-28.6), with 3.3% heavy intensity infection. School-age children in public schools; AOR: 3.92, (95% CI: 2.33-6.60), drinking river

**Data availability statement:** All relevant data supporting the findings of this study are included within the manuscript and its supporting information files.

**Funding:** The author(s) received no specific funding for this work.

**Competing interests:** The authors have declared that no competing interests exist.

water; AOR: 1.79, (95% CI: 1.08-2.98), irregular handwashing before meals; AOR: 3.18, (95% CI: 1.24-8.35), eating unwashed fruits; AOR: 2.47, (95% CI: 1.56-3.92), and habits of soil contact; AOR: 2.48, (95% CI: 1.69-3.62), were associated factors for soil-transmitted helminths infection, whereas river swimming habits; AOR: 3.46, (95% CI: 2.18-5.50), bathing in the river; AOR: 3.29, (95% CI: 2.18-5.50), male gender; AOR: 1.72, (95% CI: 1.15-2.58), and school-age children in public schools; AOR: 2.36, 95% CI (1.19-4.68), were predictors of *Schistosoma mansoni* infection.

## Conclusion

Despite the preventive chemotherapy that has been implemented in the study area, the prevalence of soil-transmitted helminths and *Schistosoma mansoni* infections persist at a high level. Hence, the preventive chemotherapy implementation could be revised and integrated with other control strategies for elimination.

## Author summary

Soil-transmitted helminthiasis and *Schistosoma mansoni (S.mansoni)* infections are among the most prevalent parasitic infections in children, typically occurring in communities that lack proper sanitation and hygiene. In Mizan-Aman Town of southwestern Ethiopia, despite seven years of a government deworming program, it remains highly prevalent among schoolchildren. The present study tested stool samples using the specialized laboratory techniques (Kato-Katz technique) from 615 children, revealing that 50.7% had soil transmitted worm infections, and 25.4% were infected with *S. mansoni*. While most cases showed low worm burden in the infected children, but some infected children had high worm burden. Risk factors involved in acquiring soil transmitted worm infection include going to a public school, drinking river water, not washing hands regularly, eating unwashed fruits, and frequently contact with soil. While *S. mansoni* infections were more predominant among boys and those who had the habit of river swimming or bathing in contaminated rivers. These findings outline that medication alone cannot help people fight against these diseases. While the deworming programs reduced the number of severe infections (high worm burden), the persistently high prevalence speaks loudly about the urgent need for comprehensive interventions. Thus, improvement in access to clean water and sanitation with hygiene is very crucial. Meanwhile, equal effort should go into constructing safe water facilities and hygiene education with proper toilets and hand-washing stations within the school. If these challenges addressed simultaneously, they may lead to a very significant reduction in infection and overall improvement of health and well-being in children.

## Introduction

Soil-transmitted helminths (STHs) infections and schistosomiasis (SCH) are among the prioritized Neglected Tropical Diseases (NTDs) by the World Health Organization (WHO), posing significant public health challenges globally [1]. STHs infections encompass diseases caused by four intestinal parasites: roundworm (*Ascaris lumbricoides*), whipworm (*Trichuris trichiura*), and hookworms (*Necator americanus* and *Ancylostoma duodenale*) [2]. While SCH is caused by six species of schistosomes (*S. mansoni*, *S. haematobium*, *S. japonicum*, *S. intercalatum*, *S. mekongi,* and S. *guineensis*) [3]. With *S. mansoni* prevalent in many sub-Saharan African countries, including Ethiopia [4].

Globally, in 2019, about 1.5 billion people were infected with STHs, while 236.6 million required preventive chemotherapy (PC) for SCH [5,6]. School-aged children (SAC) and preschool-aged children (Pre-SAC) are particularly vulnerable to these infections. SAC, according to WHO 2019 report, bear the highest burden, with over 762 million requiring PC for STHs and 124.4 million for SCH globally [7]. These infections have significant impacts, including micronutrient deficiency, anemia, cognitive impairment, and physical complications such as intestinal obstruction and rectal prolapse [8–10]. Heavy infection intensities are associated with growth stunting, reduced physical fitness, and impaired memory and cognition in Pre-SAC and SAC, ultimately affecting educational performance and school attendance [11,12].

The majority of SCH cases (90%) are concentrated in sub-Sahara African (SSA) countries, and many cases of STHs infection are found in African countries, including Ethiopia [13]. According to the Ethiopian Ministry of Health 2020 report, about 96.7 million and 53.3 million people were living in STHs infection and SCH endemic areas, respectively; with around 27.7 million and 16.9 million SAC at risk of STHs infection and SCH, respectively [14]. Moreover, an impact analysis on STHs prevalence and intensity conducted from 2000 to 2018 in SSA countries revealed that Ethiopia is among the top three countries with a majority of implementation units having over 20% STHs prevalence and had estimated prevalence of moderate and heavy intensity (M&HI) of STHs infection exceeding the WHO target threshold of 2% after years of PC implementation [15].

In 2001, the World Health Assembly expert committee proposed PC (benzimidazole for STHs and praziquantel for SCH) to reduce disease burden in endemic countries [16]. Consequently, in 2013, the Ethiopian Public Health Institute and Ministry of Health initiated a mapping of endemic areas to implement PC based on WHO guidelines. By 2015, high-risk (prevalence >50%) and moderate-risk (prevalence >20% and < 50%) areas began PC twice and once a year, respectively in the endemic area of the country. Similarly, for *S. mansoni* infection, high-risk (prevalence >50%), moderate-risk (prevalence >10% and < 50%), and low-risk (prevalence >1% and < 10%) areas started PC annually, biennially, and twice during primary school, respectively [17–19]. WHO recommends re-evaluating PC frequency for STHs after 5–6 years using a decision tree for treated population: that is; no PC if prevalence <2%, biennially for 2–10% prevalence, annually for 10–20%, maintaining previous frequency for 20–50%, or tripling PC if prevalence exceeds 50% after years of PC intervention [20].

Moreover, as per WHO guidelines for the elimination program of STHs and *S. mansoni* infections, the impact indicator, particularly the parasitological, including prevalence and intensity, should be assessed at various milestones outlined by WHO and national NTD control programs. This evaluation is essential for monitoring progress towards program goals and objectives [5,6,20]. Specifically, moderate and heavy intensity STH infections, along with heavy intensity SCH infections, should be reduced to less than 2% and 1%, respectively, to measure morbidity reduction after years of PC interventions in endemic regions. Additionally, the prevalence of both STHs and SCH should be reduced to transmission interruption level, indicating no reported autochthonous human cases in the endemic area [6,20].

Consequently, as per Ethiopian national control program guideline, Mizan-Aman town has started PC program implementation twice a year for STHs infection and once a year for *S. mansoni* infection since 2015, which grouped the area at high-risk districts (prevalence ≥50%) during the PC program start-up for both STHs and *S. mansoni* infections. However, even though the PC program has been being implemented aiming to reduce disease burden for span of seven years, to

date, the current prevalence and intensity of STHs and *S. mansoni* infections among SAC has not been assessed towards the program's targets. Therefore, the study aimed to determine the prevalence and intensity of both STHs and SCH toward elimination program targets after seven years of intervention.

## Methods

### Ethics statement

The study was conducted following ethical clearance obtained from the Institutional Review Board of the Jimma University Institute of Health (Ref. No. JUIRB002/14). Official permission letters were secured from the School of Medical Laboratory Sciences at Jimma University, the Mizan-Aman City Administration Health Office, and the Mizan-Aman City Administration Education Office. Written informed consent was obtained from parents or legal guardians of all participating children, and assent was sought from the children themselves to ensure their willingness to participate. Confidentiality of all participants' laboratory results was maintained throughout the study. Children who tested positive for intestinal parasites were referred to local health centers for appropriate treatment.

### Study setting and period

The study was conducted from January to February, 2022 in Mizan-Aman town, the capital of the Bench-Shako zone in Southwest Ethiopia's People Regional State. Mizan-Aman town is located 561 kilometers southwest of Ethiopia's capital, Addis Abeba. It has warm climate with mean annual temperatures range from 15.1 to 27 °C and an elevation of 1451m to 1753m above sea level. The mean annual rainfall ranges from 400 to 2000mm. The study area is located between 70° 0′ 0″ N and 35° 35′ 0″ E (CSA, 2007). The area is considered as a conducive for agriculture and human settlement. The major economic activity of the urban inhabitants is trading, while subsistence farming is for the surrounding rural population.

The total population of Mizan-Aman town is estimated to be 123,005, of which 63,963 are males as obtained from Mizan-Aman city administration plan commission. In the year 2021/2022 academic year, a total of 19021 children (9462 males and 9559 females) were enrolled in registered and functioning 25 primary schools (13 public schools and 12 private schools) in the town.

### Study design and population

A facility-based cross-sectional study was employed among school age children in primary schools of Mizan-Aman town during the study period. All SAC aged between 5 years to 14 years, who were present at school during the study period and able to provide an assent were included in the study.

### Sample size calculation and sampling approach

Sample size was determined using the single population proportion formula $n = \frac{Z^2 * p(1-p)}{d^2}$, based on the prevalence of the previous study from Mizan-Aman town (73.9%) [21]. The minimum sample size required for analysis was calculated using a 95% confidence level with a 5% margin of error. Additionally, a 10% non-response rate and design effect of 2 were calculated, resulting in a final sample size of 650.

To select study participants, a multistage sampling technique was used; 10 primary schools were chosen from the six kebeles; six public primary schools and four private primary schools were chosen by lottery method from each kebele. After the selection of schools, the sample size was proportionally allocated to each selected school. Accordingly, 77, 18, 122, 26, 52, 75, 50, 160, 45, ad 25 samples were allocated for Saluta, Salam, Aman, Tsinseta Mariyam, Ediget Behibret, Mizan number two, Mizan Misagana, Mizan number one, Abune teklehymanot and Gacheb number one primary schools, respectively. To select each study participant, a systematic sampling technique was applied using class roster as a

sampling frame. The k$^{th}$ interval to select the sample was every 21$^{th}$ (twentieth). For using sampling frame for each school, all class rosters (from grade one to eight) were added up as single sampling frame consecutively. In the case of participant absenteeism or not illegible, the next student was selected from class roster list.

### Questionnaire survey

A pretested questionnaire written in English and translated into Amharic was used to collect sociodemographic data and associated factors of STHs and SCH from each study participant. The data collection was carried out by a trained health professional.

### Stool collection and processing

A stool specimen of about 5 g (thumb-sized) was collected by clean, dry, wide-mouth, and leak-proof container from each of the SAC. All collected stool samples were transported to the Mizan-Tepi University Laboratory by using cold box and processed by the two-slide Kato-Katz technique (41.7mg type) [22]. After the preparation of thick smear of two-slide Kato-Katz by using WHO, 2019 SOP, all smears were kept at least for one hour at room temperature to clear the fecal material prior to examination and within 24 hours of preparation. However, for Hookworm eggs the results were read after 30 minutes of smear preparation. The examination was done by two senior laboratory technologists for each specimen and in the case of result discrepancy, the slides were reexamined by the third experienced laboratory technologist. To determine the egg per gram of stool in each Kato-Katz slide, the total counted parasite per slide was multiplied by 24 for each slide. The intensity of STHs and *S. mansoni* infection was determined according to WHO intensity classes (light, moderate and heavy intensity) [23]. Moreover, the M&HI of STHs infection was calculated according to the WHO recommended formula to calculate M&HI, as stated below [20].

$$M\&HI = \frac{Moderate\ intensity + Heavy\ intensity}{Total\ sample\ screened}$$

### Data management and analysis

Data were entered into EpiData version 4.6.06 and exported to SPSS version 26.0 for analysis after ensuring completeness. Descriptive statistics, including frequencies, bar graphs, and proportions, were used to summarize the data. Bivariate and multivariable logistic regression analyses were performed to identify associations between variables. Independent variables with a p-value less than 0.25 in the bivariate analysis were included in the multivariable logistic regression model. Adjusted odds ratios (AORs) with 95% confidence intervals (CIs) were calculated, and variables with a p-value less than 0.05 in the multivariable logistic regression were identified as significant determinants of the outcome variables.

### Operational definitions

**Parasitological indicators:** measurable reduction of prevalence and intensity of STH and *S. mansoni* infection after years of PC implementation.

   **WHO decision tree** for PC frequency after 5−6 years Kato Katz evaluation of STH prevalence

1. If the STH prevalence is ≥ 50%, the area may require PC **three times** a year.

2. If the prevalence is ≥ 20% and < 50%, the area may require PC **twice a year.**

3. If the prevalence is < 20%, the area may require PC **once a year**.

4. If the prevalence is ≥ 2% and < 10%, the area may require PC once every two years.

## Results

### Sociodemographic characteristics of the study participants

In this cross-sectional study, only 615 SAC had participated out of 650 SAC selected from Mizan-Aman primary schools, yielding a response rate of 94.6%. Of these 615 participants, 290 (47.2%) and 315 (52.8%) were males and females, respectively. Most of the study participants 411 (66.8%) were in the age range of 10–14 years. The mean age of the participants was 10.7. Majority of SAC 482 (78.2%) were from public schools, and 338 (55%) were in the 5–8 grade category, Table 1.

### Prevalence of STHs and *S. mansoni* after seven years of PC intervention

Out of 615 screened SAC, the overall prevalence of intestinal parasite was 367 (59.8%). The STHs infection was 312 (50.7%); 95% CI: (46.8-54.8); and the most frequently observed STHs species was *T. trichiura* (48%; 95% CI: 44–52), followed by *A. lumbricoides* (8.3%; 95% CI: 6.2-10.8) and hookworm (1.5%; 95% CI: 0.7-2.8). The prevalence of STHs infections was varied among schools, being highest in Gacheb primary school (87.5%), followed by Idiget behibrat primary school (77.6%), and Salute primary school (72.6%). Whereas the lowest STHs infection prevalence was observed in Mizan Misgana Academy (6.4%).

On the other hand, from the total SAC screened, 156 (25.4%; 95% CI: 22.1-28.6) were infected with *S. mansoni* and its prevalence varied among schools with the highest prevalence in Gacheb primary school (54.2%) followed by Idiget behibrat primary school (42.9%) and Aman primary school (39.1%). The lowest prevalence was observed in Mizan Misgana Academy (4.3%), Table 2.

### Intensity of STHs and *S. mansoni* after seven years of PC intervention

From the 615 SAC screened for STHs, 48.3% and 2.4% were infected with light and moderate intensity, respectively, but none of the SAC were infected with heavy intensity. Most of the moderate intensities were due to *T. trichiura* accounting

**Table 1. Sociodemographic characteristics of school age children in selected primary schools of Mizan-Aman town, 2022 (n = 615).**

| Characteristics | | | Frequency | Percent |
|---|---|---|---|---|
| Age | 5-9 | | 204 | 33.2 |
| | 10-14 | | 411 | 66.8 |
| Sex | Male | | 290 | 47.2 |
| | Female | | 315 | 52.8 |
| School type | Public | | 482 | 78.4 |
| | Private | | 133 | 21.6 |
| Schools | Selam | | 17 | 2.8 |
| | Aman catholic | | 26 | 4.2 |
| | Saluta | | 73 | 11.9 |
| | Aman primary school | | 115 | 18.7 |
| | Idiget behibrat | | 49 | 8.0 |
| | Mizan number one | | 149 | 24.2 |
| | Mizan number two | | 72 | 11.7 |
| | Gacheb | | 24 | 3.9 |
| | Mizan misgana | | 47 | 7.6 |
| | Abune t/hymanaot | | 43 | 7.0 |
| Student's grades category | 1-4 | | 338 | 55 |
| | 5-8 | | 277 | 45 |

**Table 2. Prevalence of any intestinal helminthic infections by their species among school age children at Mizan-Amman town, 2022 (n = 615).**

| Characteristics | Category | Total examined children | Prevalence of all IH | STHs infected | *S. mansoni* infected | Other IH infected |
|---|---|---|---|---|---|---|
| | | n (%) | n (%) | n (%) | n (%) | n (%) |
| Age group in years | 5-9 | 204 (33.2) | 120 (58.8) | 108 (52.9) | 38 (18.6) | 10 (4.9) |
| | 10-14 | 411 (66.8) | 247 (60.1) | 204 (49.6) | 118 (28.7) | 14 (3.4) |
| | **Total** | **615 (100)** | **367 (59.7)** | **312 (50.7)** | **156(25.4)** | **24 (3.9)** |
| Sex | Male | 290 (47.2) | 194 (66.9) | 160 (55.2) | 91 (31.4) | 12 (4.2) |
| | Female | 325 (52.8) | 173 (53.2) | 152 (46.8) | 65 (20.0) | 12 (3.7) |
| | **Total** | **615 (100)** | **367 (59.7)** | **312 (50.7)** | **156(25.4)** | **24 (3.9)** |
| School type | Public | 482 (78.4) | 331 (68.7) | 286 (59.3) | 144 (29.9) | 20 (4.1) |
| | Private | 133 (21.6) | 36 (27.1) | 26 (19.5) | 12 (9.0) | 4 (3.0) |
| Schools | Selam | 17 (2.8) | 12 (70.6) | 8 (47.1) | 4 (23.5) | 3 (17.6) |
| | Aman catholic | 26 (4.2) | 8 (30.7) | 7 (26.9) | 4 (15.4) | 0 (0) |
| | Saluta | 73 (11.9) | 55 (75.4) | 53 (72.6) | 15 (20.5) | 2 (2.7) |
| | Aman primary school | 115 (18.7) | 70 (60.9) | 57 (49.6) | 45 (39.1) | 3 (2.6) |
| | Idiget behirat | 49 (8.0) | 39 (79.6) | 38 (77.6) | 21 (42.9) | 3 (6.1) |
| | Mizan number one | 149 (24.2) | 95 (63.8) | 72 (48.3) | 42 (28.2) | 9 (6.0) |
| | Mizan number two | 72 (11.7) | 50 (69.4) | 45 (62.5) | 8 (11.1) | 3 (4.2) |
| | Gacheb | 24 (3.9) | 22 (91.7) | 21 (87.5) | 13 (54.2) | 0 (0) |
| | Mizan misgana | 47 (7.6) | 5 (10.6) | 3 (6.4) | 2 (4.3) | 0 (0) |
| | Abune t/hymanaot | 43 (7.0) | 11 (25.6) | 8 (18.6) | 2 (4.7) | 1 (2.3) |
| | **Total** | **615 (100)** | **367 *(59.7)*** | **312 *(50.7)*** | **156 *(25.4)*** | **24 *(3.9)*** |

IH: intestinal helminths.

for 2% (representing, 83.3% of the total 2.4% moderate intensity of overall STH). The M&HI of overall STH was 2.4%. On the other hand, from the 615 SAC screened for *S. mansoni*, 12.4% of them were infected with light intensity, 9.8% with moderate intensity, and 3.3% with heavy intensity, Table 3.

## Prevalence of helminthic multiple infections

In this study, 135 (21.95%) of the screened SAC were found to have multiple infections, (i.e., double or triple infection) involving combinations of STHs with STHs and/or with *S. mansoni*), and 37.50% was with mono infections Fig 1.

**Table 3. Prevalence and intensity of STHs and S. mansoni infection among SAC in primary schools of Mizan Aman town, 2022 (n = 615).**

| Parasite species | Number of infected children n (%) | Intensity of infection | | |
|---|---|---|---|---|
| | | Light n (%) | Moderate n (%) | Heavy n (%) |
| *T. trichiura* | 295 (48) | 283 (46.0) | 12 (2.0) | 0 |
| *A. lumbricoides* | 51 (8.3) | 48 (7.8) | 3 (0.5) | 0 |
| *H. worm* | 9 (1.5) | 9 (1.5) | 0 | 0 |
| Any STHs species | 312 (50.7) | 297 (48.3) | 15 (2.4) | 0 |
| *S. mansoni* | 156(25.4) | 76 (12.4) | 60 (9.8) | 20(3.3) |

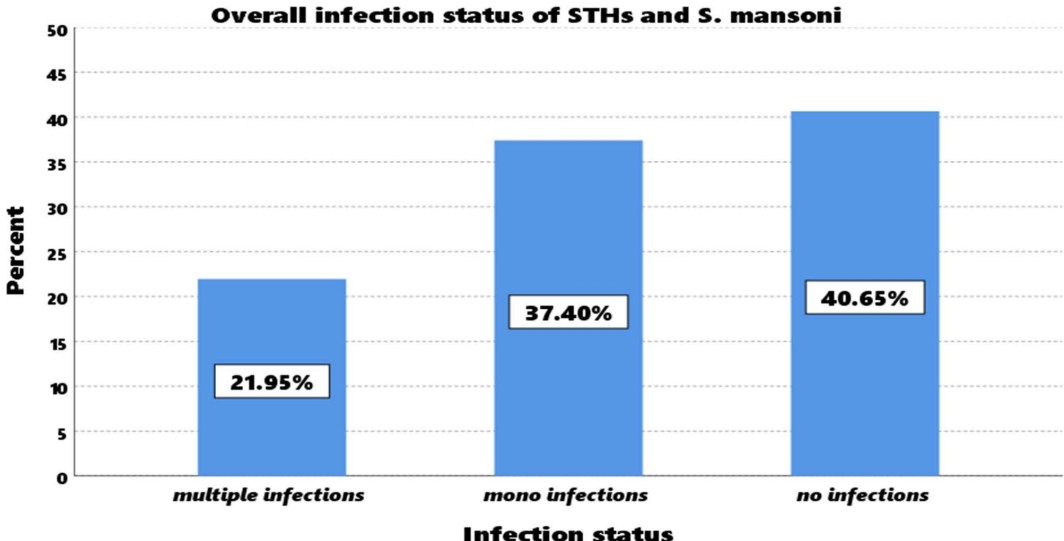

**Fig 1. Overall multiple infections and mono-infections of STHs and *Schistosoma mansoni*.** This figure illustrates the distribution of mono-infections and multiple infection among SAC, indicating the prevalence of each parasitic infection.

### Factors associated with soil-transmitted helminths infection

Multivariable logistic regression analysis output indicate that; SAC attending public schools; AOR: 3.92, 95% CI (2.33-6.60), P < 0.001, irregular washing of hands before meal, AOR: 3.18, 95% CI (1.24-8.35), P = 0.016, habit of soil contact with hand, AOR: 2.48, 95% CI (1.69-3.62), P < 0.001, using river/spring water as drinking source; AOR: 1.79, 95% CI (1.08-2.98), P < 0.025, and habit of eating unwashed fruits; AOR: 2.47, 95% CI (1.56-3.92), P=<0.001 had statistically significant association with STHs infection. However, other variables such as sex, finger trimming, and finger biting habits were not significantly associated with STHs infection in the current study, Table 4.

### Factors associated with *Schistosoma mansoni* infection

After multivariable logistic regression analysis; SAC attending public school; AOR: 2.36, 95% CI (1.19-4.68), P = 0.014; male gander; AOR: 1.72, 95% CI (1.15-2.58), P = 0.008; river bathing or washing habit; AOR: 3.29, 95% CI (2.18-5.50), P < 0.001, and river swimming habit; AOR: 3.46, 95% CI (2.18-5.50), P < 0.001 had statistically significant association with *S. mansoni* infection. But other variables such as age group, a nearby river around the home, and barefoot river crossing were not significantly associated with *S. mansoni* infection in the current study, Table 5.

### Discussion

STHs and *S. mansoni* infections are classified as NTDs and continue to pose significant public health challenges in various regions globally, particularly in low-resource settings [7]. In light of this, different endemic countries, including Ethiopia, have been undertaking control activities among SAC such as PC intervention in collaboration with WHO for some years [16,18]. Hence, assessing the parasitological indicators (prevalence and intensity) is a very crucial to determine the program's progress in areas where the elimination program is undergoing [24].

The STHs infection prevalence after seven years of implementing PC in the present study area was 50.7%, which is in agreement with the finding of a similar study conducted in southern Ethiopia (54.7%) [25]. However, it is notably lower than the prevalence observed in a similar study conducted in Rwanda after ten years of implementing PC (77.7%) [26].

**Table 4. Bivariate and multivariable logistic regression showing factors associated with STHs infection among school age children in selected primary schools of Mizan-Aman town, 2022 (n = 615).**

| Variables | | Kato-Katz result | | Bivariate LR | | Multivariable LR | |
|---|---|---|---|---|---|---|---|
| | | For STHs | | COR (95% CI) | P value | AOR (95% CI) | P value |
| | | Yes n (%) | No n (%) | | | | |
| Sex | Male | 160 (55.2) | 130 (44.8) | 1.40 (1.02-1.93) | 0.038 | 1.34 (0.94-1.92) | 0.107 |
| | Female | 152 (46.8) | 173 (53.2) | 1 | | 1 | |
| School type | Public | 286 (59.3) | 196 (40.7) | 6.00 (3.8-9.6) | <.001 | 3.92 (2.33-6.60) | <0.001* |
| | Private | 26 (19.5) | 107 (80.5) | 1 | | 1 | |
| Frequency of hand wash before meal | Always | 290 (49.4) | 297 (50.6) | 1 | | 1 | |
| | Sometimes | 22 (78.6) | 6 (21.4) | 3.76 (1.50-9.40) | 0.005 | 3.18 (1.24-8.35) | 0.016* |
| Trimmed finger nail | Yes | 143 (46.1) | 162 (53.1) | 1 | | 1 | |
| | No | 169 (54.5) | 141 (45.5) | 1.36 (0.99-1.86) | 0.059 | 0.69 (0.47-1.02) | 0.062 |
| Finger nail biting habit | Yes | 187 (55.5) | 150 (44.5) | 1.53 (1.11-2.00) | 0.009 | 1.40 (0.97-2.02) | 0.071 |
| | No | 125 (44.9) | 153 (55.1) | 1 | | 1 | |
| Habit of soil contact with hand | Yes | 192 (69.1) | 120 (30.9) | 2.44 (1.76-3.37) | <.001 | 2.48 (1.69-3.62) | <0.001* |
| | No | 120 (39.6) | 183 (60.4) | 1 | | 1 | |
| Source drinking water | Tap | 213 (45.0) | 260 (55.0) | 1 | | 1 | |
| | Well | 26 (63.4) | 15 (36.6) | 2.12 (1.09-4.10) | 0.026 | 1.19 (0.59-2.40) | 0.633 |
| | Spring/River | 73 (72.3) | 28 (27.7) | 3.18 (1.99-5.10) | <.001 | 1.79 (1.08-2.98) | 0.025* |
| Washing fruits before eating | Yes | 214 (44.7) | 265 (55.3) | 1 | | 1 | |
| | No | 98 (72.1) | 38 (27.9) | 3.19 (2.11-4.84) | <.001 | 2.47 (1.56-3.92) | <0.001* |

*Indicates the significant variables, COR = crude odds ratio, AOR = adjusted odds ratio, 1 = reference group, LR = logistic regression.

This difference in prevalence might be due to the difference in effectiveness of PC implementation, like the percentage of PC coverage, regularity of PC implementation, strains of STHs species in the area, and/or the difference in the baseline prevalence of STHs infection during the startup of PC intervention in the area.

On the other hand, it appears that, despite the implementation of a PC program over a span of seven years in the study area, the current prevalence of STHs infection is significantly not on track of achieving target set by the WHO for 2030 and that of Ethiopia's third national strategic plan targets for NTDs control. The target aims to achieve less than 2% prevalence of STHs infection in endemic areas by 2030 [14,20].

Moreover, according to WHO decision tree, one of the WHO's targets is to reduce the quantity of tablets needed for PC by lowering the STHs prevalence after 5–6 years of implementation. However, unfortunately, the current high prevalence of STHs infection categorizes the area under the designation of requiring PC three times a year (if the prevalence exceeds 50%) for the treated population in STHs endemic areas. Consequently, this situation may demand an increment in the number of tablets distributed in the area [6,20]. This inconsistent finding with WHO targets in the present study area after years of PC implementation might be due to irregular PC implementation, like an interruption of PC implementation in the study area during 2020 and 2021 due to the Coronavirus Disease 2019 (COVID-19) pandemic, the absence of complementary WASH (Water, Sanitation, and Hygiene) interventions alongside PC, a deficiency in coordinated health education efforts in the study area as noted by Bech-Sheko Zonal health office and Mizan-Aman health office, and/or might be challenges related to drug efficacy.

The prevalence of *S. mansoni* infection in the present study area after seven years of PC program implementation was 25.4%, which is consistent with what was reported from a similar study conducted in southern Ethiopia, 25.8% [25]. This finding, however, is lower than a similar study conducted in the same study area nine years ago prior to PC intervention

**Table 5. Bivariate and multivariable logistic regression showing factors associated with S. mansoni infection among school age children in selected primary schools of Mizan-Aman town, 2022 (n = 615).**

| Variables | | Kato-Katz result | | Bivariate LR | | Multivariable LR | |
|---|---|---|---|---|---|---|---|
| | | *For S. mansoni* | | COR (95% CI) | P Value | AOR (95% CI) | P Value |
| | | Yes, n (%) | No, n (%) | | | | |
| Age group in years | 5-9 | 38 (**18.6**) | 166 (**81.4**) | 1 | | 1 | |
| | 10-14 | 118 (**28.7**) | 293 (**71.3**) | **1.76** (1.17-2.66) | **0.007** | **1.56** (.0.99-2.45) | **0.051** |
| Sex | Male | 91 (**31.4**) | 199 (**68.6**) | **1.83** (1.26-2.64) | **0.001** | **1.72** (1.15-2.58) | **0.008**[*] |
| | Female | 65 (**20.6**) | 250 (**79.4**) | 1 | | 1 | |
| School type | Public | 144 (**29.9**) | 338 (**70.1**) | **4.30** (2.30-8.02) | **<0.001** | **2.36** (1.19-4.68) | **0.014**[*] |
| | Private | 12 (**9**) | 121 (**91**) | 1 | | 1 | |
| Living nearby river | Yes | 96 (**30.5**) | 219 (**69.5**) | **1.75** (1.21-2.54) | **0.003** | **1.26** (0.83-1.92) | **0.271** |
| | No | 60 (**20**) | 240 (**80**) | 1 | | 1 | |
| Bare foot River crossing | Yes | 98 (**30.8**) | 220 (**69.2**) | **1.84** (1.26-2.66) | **0.001** | **0.93** (0.61-1.43) | **0.740** |
| | No | 58 (**19.5**) | 239 (**80.5**) | 1 | | 1 | |
| Bathing/ washing in the river | Yes | 123 (**37.4**) | 206 (**62.6**) | **4.58** (2.99-7.01) | **<0.001** | **3.29** (2.07-5.23) | **<0.001**[*] |
| | No | 33 (**11.5**) | 253 (**88.5**) | 1 | | 1 | |
| Swimming in the river | Yes | 126 (**36.8**) | 216 (**63.2**) | **4.76** (3.05-7.32) | **<0.001** | **3.46** (2.18-5.50) | **<0.001**[*] |
| | No | 30 (**10.9**) | 243 (**89.1**) | 1 | | 1 | |

[*]Indicates the significant variables, **COR** = crude odds ratio, **AOR** = adjusted odds ratio, **1** = reference group, **LR** = logistic regression.

[21]. This reduction in prevalence might be due to PC implementation in the area, the difference in sampling techniques, and/or the difference in the study time.

On the other hand, the WHO guidelines for the control and elimination of human schistosomiasis classify endemic areas into three classes based on their prevalence, accordingly; in the present study, the prevalence of *S. mansoni* infection remained at a moderate prevalence of 25.4% despite the PC intervention having been given over the span seven years in the area [27]. Moreover, according to the WHO 2030 human schistosomiasis control guideline, this prevalence classifies the area under the category of once-a-year PC intervention [5].

Moreover, according to recommendation number six of the WHO guideline on control and elimination of human schistosomiasis and Ethiopia's third NTDs strategic plan, the prevalence identified in the current study indicates a significant gap from achieving transmission interruption (defined as having no autochthonous human cases reported in the area). Consequently, much more effort is expected from the concerned body to bring down the prevalence of the study area to interruption of transmission level [5,14]. This continuing *S. mansoni* infection transmission might be due to irregular PC implementation, like the interruption in 2020 due to COVID-19 pandemic, the absence of the WASH program, the absence of coordinated health education in the study area as reported by the Bech-Sheko Zonal health office, and Mizan-Aman health office, and/or challenges related to drug efficacy.

The present study finding of moderate and heavy intensity (M&HI) of STHs at 2.4% is slightly surpasses WHO 2030 target number one and Ethiopia's third NTDs strategic plan targets, which indicate that elimination of STHs infection as a public health problem is achieved at less than 2% M&HI after five years of PC implementation, and WHO planned that 60% of endemic countries would achieve this less than 2% M&HI by the 2023 milestone [6,14,20]. But the current finding highlighted the fact that, if PC intervention is effectively continued without interruption in the area, the current status is very promising to achieve elimination of STHs infection as a public health problem (<2% M&HI) by the second milestone, which is that 70% of endemic areas should achieve <2% M&HI by the year 2025 [6]. This difference with the first WHO milestone target (2023) might be due to the remaining round of the PC program, irregular PC implementation, particularly

interruptions during 2020 and 2021 as reported by Bech-Sheko Zonal health office and Mizan-Aman health office, and/or challenges related to drug efficacy.

The present *S. mansoni* heavy intensity of 3.3% is in agreement with what was suggested in the WHO schistosomiasis control program strategic plan of 2011–2020, which was that endemic areas implementing PC intervention program should achieve schistosomiasis morbidity control (<% 5 heavy intensity of *S. mansoni*) after conducting 5–10 years of PC intervention [27]. However, still, more activity is expected from this area to achieve the WHO 2030 NTD control program, which states that by 2023, 63% of endemic areas that began PC intervention will have eliminated SCH as a public health problem (<1% of heavy intensity) [6], as well as Ethiopia's third NTDs strategic plan target of reducing *S. mansoni* heavy intensity to < 1% by 2023 [14]. This inconsistency about elimination of SCH as a public health problem with WHO and Ethiopia's third NTDs strategic plan target might be due to the interruption of PC in the study area during 2020 as reported by Bech-Sheko Zonal health office and Mizan-Aman health office, the remaining round of PC intervention and/or challenges related to drug efficacy.

Although, the elimination of SCH as public health was not achieved at the time of the current study, the fact that the present *S. mansoni* heavy intensity of 3.3% which is closest to the cutoff value (<1% heavy infection) is very promising [27]. So, if PC intervention is effectively continued without interruption in the area or, especially, if tailored to WASH program activities, it might achieve the elimination of *S. mansoni* infection as a public health problem by the next milestone, which is by 2025 [27].

On the other hand, the present study also revealed that, even if the PC intervention has been given over span seven years, unfortunately, the prevalence of *T. trichiura* compared to other STHs species remained high (48%). This predominating prevalence of *T. trichiura* compared to other STHs species was also observed in a similar study conducted in Rwanda after ten years of PC program implementation [26]. As indicated in several studies conducted elsewhere, this high prevalence of *T. trichiura* might be related to the low efficacy of the current drug regimen in use for PC intervention, i.e., the administration of single albendazole, which has low efficacy compared to triple dose administration, which might result in a high prevalence of *T. trichiura* after years of intervention due to its low cure rate [28–33]. Although its practical implementation remains a challenge in PC program intervention, using multiple doses for consecutive days or combination therapy might increase the efficacy against *T. trichiura*, [32,33].

Moreover, one of the current challenges arising in PC intervention is the emergence of an albendazole-resistant *T. trichiura* strain with the mutant beta-tubulin gene in different endemic areas, which could impose problems in the reduction of the prevalence of this species when using albendazole [33–36]. So, although molecular investigation is needed to confirm the existence of resistant strains in the present study area, the current high prevalence of *T. trichiura* might be related to the existence of this mutant beta-tubulin gene strain in the area.

It is also alarming that the prevalence of *T. trichiura* is higher than the reported prevalence (18.7%) from the same study area conducted in 2013 before PC intervention [21]. This difference in prevalence might be due to differences in sample size, study time, sampling techniques, using double slide Kato-Katz in the present study, and/or a drug-resistant strain of *T. trichiura*.

From the total screened SAC, the overall prevalence of multiple infections (double or triple infection) of *S. mansoni* and STHs species was 21.95%. According to various scientific studies, morbidity is not solely linked to moderate and/or heavy intensity helminth infections; multiple helminth infections also significantly contribute to the development of various morbidities such as anemia, malnutrition, and stunting in infected individuals [37–41]. So, when developing an STHs infection and schistosomiasis control strategy, due consideration should be given to multiple infections. Because the high multiple infections rate might be due to conducive environmental conditions for the development and survival of parasitic stages, a lack of clean and treated water supply, poor hygienic standards, and low socioeconomic conditions.

The present study finding also revealed that different risk factors were associated with the prevalence of STHs and *S. mansoni* infections in the study area. For instance, SAC in public schools were more vulnerable to STHs infection than in

private schools. This finding is in agreement with findings previously reported from Mizan-Aman Town [21], Jimma Town [42], and Nigeria [43]. This high risk among SAC in a public school might be due to differences in socioeconomic status, sanitation around the residence, and sanitation facilities in the school itself.

The STHs infection was also significantly associated with the frequency of handwashing before meals; individuals who did not regularly wash their hands were at a higher risk compared to those who practiced regular handwashing. This finding is in agreement with what was reported from a study conducted in Sigmo, Hawassa, Butajira Town, southwest Shewa, and Afghanistan [44–48]. This could be due to the lack of hand washing before meals, may increase the likelihood of acquiring parasites from contaminated hands carrying infective stages during feeding. Similarly, children who had a habit of contacting soil with their hands were more at risk than their counterparts. This finding is also supported by a study conducted in Butajira Town and Southwest Shewa [47,48]. Given that STHs are transmitted to humans through contaminated soil containing feces defecated in an open environment, playing with bare hands on the soil might increase the risk of infection.

Furthermore, STH infection was significantly linked to the practice of not washing fruits before consumption; children who had no habit of washing fruits before eating were more at risk than their counterparts. This finding is consistent with what was previously reported from Mizan-Aman Town [21]. This high risk could be due to the fact that fruits might be contaminated with soil containing infective stages during their harvesting and selling time in the markets [49]. According to the finding reported in Jimma Town, vegetables and fruits were highly contaminated with STHs species [50]. Additionally, STHs infection was significantly associated with using of springs or rivers as sources of drinking water, which is in agreement with the results reported from Mizan-Aman Town, southwest Shewa, and south-eastern Nigeria [21,48,51]. This could be due to springs or rivers waters are open to be contaminated by human faeces defecated in the open environment.

The present study also revealed that SAC in public schools were more vulnerable to *S. mansoni* infection than SAC in private schools. This finding is in agreement with a previous finding reported from Mizan-Aman Town [21]. This high risk of *S. mansoni* infection among SAC enrolled in public schools might be due to the differences in exposure to rivers for swimming, washing, bathing, and/or fetching because most of SAC enrolled in public school has low socio-economic status when compared with SAC in private schools, which increase the contact frequency of infested water with cercaria.

Moreover, this current study also showed that males were more likely to be infected with a *S. mansoni* infection than females. This finding is in agreement with findings previously reported from Jimma Town, around Lake Tana, northwestern Ethiopia, and in northern Ghana [52–54]. This could be due to the fact that males might spend more time doing outdoor activities, which might increase the chance of frequently contacting the infested water with cercaria. Similarly, washing or bathing in the rivers, and swimming in the rivers were significantly associated with *S. mansoni* infection, as indicated in some articles; these risk factors might be due to the fact that contact with the river is the only route of *S. mansoni* transmission [55–57]. This finding is in agreement with what was previously reported from a study conducted in the current study area, around Lake Tana, in northwestern Ethiopia, and in northern Ghana [21,53,54]. A limitation of this study is its cross-sectional design, which restricts our ability to establish causal relationships between variables. Since data were collected at a single point in time, we cannot determine whether the exposures preceded the outcomes. Future longitudinal studies are needed to confirm these associations and provide a deeper understanding of the causal mechanisms involved.

## Conclusion

The study found that despite seven years of PC intervention in the area, both STHs and *S. mansoni* infections remained prevalent among SAC at high rates, failing to meet WHO elimination targets. Particularly, the prevalence of *T. trichiura* was disproportionately high, contrary to WHO targets. Similarly, the intensity of both STHs and *S. mansoni* infections did not meet the WHO target of elimination as public health problems. Finally, in the present study, STHs infections were

associated with factors such as school type, water sources, handwashing habits, and soil contact, while *S. mansoni* infection was linked to river activities, gender, and school type. Thus, to achieve the elimination of STHs and *S. mansoni* infections in the present study area, all concerned body should integrate PC intervention with all other WHO 2030 recommended strategies, such as, implementation of the WASH program [5,20], coordinated promotion of health education for the community [20], snail intermediate host control [5], and expanding PC to all at-risk populations for both STHs infection and SCH [5,20]. Moreover, to uncover the underlying factors contributing to the high prevalence of *T. trichiura*, researchers and other concerned body could investigate drug efficacy through various regimens or combination therapies against *T. trichiura* and molecular investigation for the presence or absence of albendazole-resistant strains of *T. trichiura* in the area.

## Supporting information

**S1 File. SPSS dataset template.** This file includes the data structure and variable definitions for the SPSS dataset template that was utilized for statistical analysis in this study.
(SAV)

## Author contributions

**Conceptualization:** Mitiku Abera, Tariku Belay, Daniel Emana, Zeleke Mekonnen.

**Data curation:** Mitiku Abera, Daniel Emana, Zeleke Mekonnen.

**Formal analysis:** Mitiku Abera, Zeleke Mekonnen.

**Funding acquisition:** Tariku Belay, Daniel Emana.

**Investigation:** Mitiku Abera.

**Methodology:** Mitiku Abera, Zeleke Mekonnen.

**Project administration:** Tariku Belay, Daniel Emana.

**Software:** Mitiku Abera.

**Supervision:** Tariku Belay, Zeleke Mekonnen.

**Validation:** Mitiku Abera, Tariku Belay, Daniel Emana, Zeleke Mekonnen.

**Visualization:** Mitiku Abera, Tariku Belay, Daniel Emana, Zeleke Mekonnen.

**Writing – original draft:** Mitiku Abera, Daniel Emana.

**Writing – review & editing:** Mitiku Abera, Zeleke Mekonnen.

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
