## [Decision Letter · Decision Letter 0]

23 Jan 2025

PNTD-D-24-01726

Assessing the Status of Soil-transmitted Helminths and Schistosoma mansoni infections toward the parasitological indicators set by the elimination program after seven years of preventive chemotherapy  among school-age children in Mizan-Aman town

Dear Dr. Abera,

Thank you for submitting your manuscript to PLOS Neglected Tropical Diseases. After careful consideration, we feel that it has merit but does not fully meet PLOS Neglected Tropical Diseases's publication criteria as it currently stands. Therefore, we invite you to submit a revised version of the manuscript that addresses the points raised during the review process.

Please submit your revised manuscript within 60 days Mar 24 2025 11:59PM. If you will need more time than this to complete your revisions, please reply to this message or contact the journal office at plosntds@plos.org. Please include the following items when submitting your revised manuscript:

We look forward to receiving your revised manuscript.

Kind regards,

Uwem Friday Ekpo, PhD

Academic Editor

Jong-Yil Chai

Section Editor

Shaden Kamhawi

co-Editor-in-Chief

Paul Brindley

co-Editor-in-Chief

**Journal Requirements:**

1) Please ensure that the Title in your manuscript file and the Title provided in your online submission form are the same.

3) We note that your Data Availability Statement is currently as follows: "The data supporting our findings are incorporated within the manuscript and. Additional data supporting the study are also available from the corresponding author upon request". Please confirm at this time whether or not your submission contains all raw data required to replicate the results of your study. Authors must share the “minimal data set” for their submission. PLOS defines the minimal data set to consist of the data required to replicate all study findings reported in the article, as well as related metadata and methods (https://journals.plos.org/plosone/s/data-availability#loc-minimal-data-set-definition).

- The points extracted from images for analysis..

**Reviewers' Comments:**

Reviewer's Responses to Questions

**Key Review Criteria Required for Acceptance?**

**Methods**

-Are the objectives of the study clearly articulated with a clear testable hypothesis stated?

-Is the study design appropriate to address the stated objectives?

-Is the population clearly described and appropriate for the hypothesis being tested?

-Is the sample size sufficient to ensure adequate power to address the hypothesis being tested?

-Were correct statistical analysis used to support conclusions?

-Are there concerns about ethical or regulatory requirements being met?

Reviewer #1: -Are the objectives of the study clearly articulated with a clear testable hypothesis stated?

- Yes the manuscript have clear objectives to assess the impact of multiple intervention of S.haematobium prevalence and further assess the factors contributing to persistence transmission

-Is the study design appropriate to address the stated objectives?

Yes, the study design is appropriate

-Is the population clearly described and appropriate for the hypothesis being tested?

-The study population is well described for schools and community

-Is the sample size sufficient to ensure adequate power to address the hypothesis being tested?

- The sample size is well described and the calculations are clear with power to test the hypothesis

-Were correct statistical analysis used to support conclusions?

-Statistics are clear

-Are there concerns about ethical or regulatory requirements being met?

- Ethical standards were followed.

Reviewer #2: (No Response)

**Results**

-Does the analysis presented match the analysis plan?

-Are the results clearly and completely presented?

-Are the figures (Tables, Images) of sufficient quality for clarity?

Reviewer #1: -Does the analysis presented match the analysis plan?

- The analysis matches with the analysis plan

-Are the results clearly and completely presented?

- The results match the study objectives

-Are the figures (Tables, Images) of sufficient quality for clarity?

- All are clear

Reviewer #2: (No Response)

**Conclusions**

-Are the conclusions supported by the data presented?

-Are the limitations of analysis clearly described?

-Do the authors discuss how these data can be helpful to advance our understanding of the topic under study?

-Is public health relevance addressed?

Reviewer #1: The conclusion is supported by the results

-Study limitations not described and no nay limitations on the analysis described

- Yes, the discussion point out the need to address poverty which is the main central point for persistent transmission of S.haematobium infection

-Yes, very relevant manuscript and data are relevant to show the impact of multiple interventions

Reviewer #2: (No Response)

**Editorial and Data Presentation Modifications?**

Reviewer #1: No comments

Reviewer #2: (No Response)

**Summary and General Comments**

Reviewer #1: - The manuscript present very good data from endemic areas though the impact of the interventions was not that much as expected to the effort which went into data collection, indicating that schistosomiasis remain as a complex disease and there is a need to address the central point, the poverty. It will be interesting to access the treatment coverage in these IU.

Minor comments. No limitations were indicated which are obvious, from the study design and the uptake of the treatment (coverage and compliance) were not reported

Line 29- there is no large microhaematuria- this is Macrohaematuria, the same for line 30'

50-51- need english grammar check

229-0238- repetition of results, not needed

Reviewer #2: (No Response)

PLOS authors have the option to publish the peer review history of their article (what does this mean? ). If published, this will include your full peer review and any attached files.

**Do you want your identity to be public for this peer review?** For information about this choice, including consent withdrawal, please see our Privacy Policy .

Reviewer #1: No

Reviewer #2: **Yes: ** Jean T. Coulibaly

**Figure resubmission:**

**Reproducibility:**



---

## [Editor Report · Decision Letter 1]

14 Apr 2025

Dear Mr. Abera,

We are pleased to inform you that your manuscript 'Assessing Soil-Transmitted Helminths and Schistosoma mansoni Infections Using Parasitological Indicators After Seven Years of Preventive Chemotherapy Among School-Age Children in Mizan-Aman Town' has been provisionally accepted for publication in PLOS Neglected Tropical Diseases.

Best regards,

Jong-Yil Chai

Section Editor

Jong-Yil Chai

Section Editor

Shaden Kamhawi

co-Editor-in-Chief

Paul Brindley

co-Editor-in-Chief

The revised version of this manuscript is acceptable for publication.

---

## [Editor Report · Acceptance letter]

Dear Mr. Abera,

We are delighted to inform you that your manuscript, "Assessing Soil-Transmitted Helminths and Schistosoma mansoni Infections Using Parasitological Indicators After Seven Years of Preventive Chemotherapy Among School-Age Children in Mizan-Aman Town," has been formally accepted for publication in PLOS Neglected Tropical Diseases.

Best regards,

Shaden Kamhawi

co-Editor-in-Chief

Paul Brindley

co-Editor-in-Chief
